# Lepton flavour violation and neutrino masses

Ana M. Teixeira⋆

Université Clermont Auvergne, CNRS/IN2P3, LPC, F-63000 Clermont-Ferrand, France

⋆ ana.teixeira@clermont.in2p3.fr

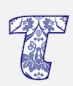 *Proceedings for the 15th International Workshop on Tau Lepton Physics, Amsterdam, The Netherlands, 24-28 September 2018*

## Abstract

**Neutrino oscillations provided the first evidence for the violation of flavour in the lepton sector, and established a clear need to consider extensions of the Standard Model. Many new phenomena can emerge from these New Physics (NP) constructions, among which processes violating lepton number and charged lepton flavour, all clear signals of New Physics. Following a short overview of the status of experimental searches, we comment on the prospects of several models of massive neutrinos, from minimal constructions to complete NP models, to the above mentioned observables.**

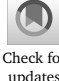 Check for updates

# 1   Introduction

As of today, strong compelling arguments suggest that the Standard Model (SM) cannot provide the ultimate description of nature. In addition to numerous theoretical issues, the SM fails to offer a viable dark matter (DM) candidate, cannot explain the the matter-antimatter asymmetry of the Universe, and lacks a mechanism capable of accounting for massive neutrinos and leptonic mixings.

Neutrino oscillations provided the first laboratory evidence for New Physics (NP), implying that the SM should be extended; moreover, the extreme smallness of their masses and their unique nature (being the only known particle which can be a Majorana fermion) further point to the interesting possibility that neutrino masses arise from a mechanism different from the one at the origin of SM fermion masses.

Several well-motivated SM extensions - relying on additional fields, extended gauge groups, or even complete New Physics frameworks - can successfully accommodate neutrino oscillation data, many also addressing the DM problem, and even providing an explanation to the baryon asymmetry of the Universe (BAU). However, and especially in view of the negative results of direct NP searches at high-energies, it remains unclear which is the SM extension at work.

Fortunately, in many of these models, $\nu$ oscillation phenomena are not the only low-energy NP signal that can be observed: massive neutrinos and flavour violation in the lepton sector open the way to many other new experimental signals. These correspond to processes that are either forbidden or extremely suppressed in the SM, including charged lepton flavour violation (cLFV), lepton number violation (LNV), contributions to lepton dipole moments, among many others. Currently, numerous high-intensity facilities, dedicated to look for these very rare processes, offer a golden laboratory to study the underlying model of NP, and ultimately shed light on the mechanism of neutrino mass generation.

In what follows, we first provide a rapid overview of the experimental status of several cLFV observables (including leptonic radiative and three-body decays, flavour violation in muonic atoms, and leptonic and semileptonic meson decays), as well as LNV observables (from neutrinoless double beta decay to rare meson and tau decays). We then discuss the potential of specific NP realisations concerning these cLFV and LNV observables, focusing on how the interplay of distinct signals might be explored to test a given SM extension: we consider minimal ad-hoc SM extensions, several seesaw realisations, and complete frameworks, as for instance supersymmetric models.

# 2   Experimental status of cLFV and LNV observables

In the original formulation of the SM, neutrinos are - by construction - massless fermions; the absence of right-handed neutral fermions and/or a Higgs triplet preclude any neutrino mass term. Moreover, having total lepton number as an (accidental) symmetry of the SM ensures that neutrinos remain massless to all orders in perturbation theory. This further implies that individual lepton flavours are strictly conserved, in both charged and neutral current interactions.

In the case of minimal SM extensions such as the $\text{SM}_{\nu_R}$, in which Dirac right-handed neutrinos are added to the SM particle content, neutrino oscillation data (squared mass differences and mixing angles) can be accounted for; charged current interactions do violate individual neutral lepton flavours, and this is parametrised via the $U_{\text{PMNS}}$ mixing matrix [1]. Charged lepton flavour violating transitions (for instance radiative $\ell_i \to \ell_j \gamma$ decays) can now occur - however, their rates are hugely suppressed by the tiny neutrino masses, lying beyond ex-

perimental reach. Thus, the observation of a cLFV transition, or the violation of total lepton number (strictly conserved for Dirac neutrinos), constitues clear evidence of New Physics, beyond the minimal $SM_{\nu_R}$.

Searches for cLFV and LNV processes are currently being carried by numerous collaborations, and several new experiments are expected to start taking data in the near future. Be it in the case of a discovery, or of an improvement in the sensitivity, the information that will be gathered clearly complements the one arising from direct searches for NP at the LHC or in future high-energy colliders (ILC, FCC, ...).

## 2.1 cLFV rare transitions and decays

In the charged lepton sector, a number of cLFV processes stems from the "muonic channels". The latter include purely leptonic decays, or then transitions associated with muonic bound states[1].

Radiative cLFV muon decays, $\mu^+ \to e^+\gamma$, have been searched since the 1940's; the current bound on these decays is BR($\mu \to e\gamma$)$< 4.2 \times 10^{-13}$, obtained by the MEG Collaboration at PSI [6]. In the future, MEG II is expected to improve the sensitivity to $6 \times 10^{-14}$ [7] (see also [8]). The three-body muon decay, $\mu^+ \to e^+e^-e^+$ also offers excellent prospects to look for cLFV. At present, the best bound is still that of SINDRUM II [9], BR($\mu \to 3e$)$< 1.0 \times 10^{-12}$, expected to be significantly improved in the coming years by the Mu3e collaboration at PSI to around $10^{-15}$ [10] (possibly $10^{-16}$, should very high-intensity muon beams be available).

Many interesting cLFV processes can be studied when muons are trapped and form the so-called "muonic atoms". When negatively charged muon beams hit a target, a muon can be stopped, and cascade down in energy until it effectively forms a 1s bound state. Other than SM allowed decays, the muonic atom can undergo a neutrinoless muon-electron conversion, $\mu^- + (A, Z) \to e^- + (A, Z)$; for spin-independent decays (coherent process)[2], the rate generaly grows with increasing atomic number, being maximal for $30 \leq Z \leq 60$ [12]. The best limit has been obtained for Gold targets, CR($\mu-e$, Au)$< 7 \times 10^{-13}$, also by the SINDRUM Collaboration [13]. In the future, several experiments will be dedicated to looking for muon-electron conversion: DeeMe [14] aims at reaching a sensitivity of $10^{-14}$ for SiC targets; working with Aluminium targets, Mu2e at Fermilab [15] expects to reach $3 \times 10^{-17}$, while at JPARC the goal of the COMET experiment is to reach $10^{-15(-17)}$ in its Phase I (II) [16].

The muonic atom can also decay into a pair of electrons, $\mu^-e^- \to e^-e^-$ [17, 18]; the associated decay rate is strongly enhanced by the Coulomb interaction between the muon and the electron wave functions, scaling with the atomic number as $(Z-1)^3$ (or even more, especially in the case of very large nuclei). This observable has not yet been experimentally searched for, but could be included in the Physics Programme of COMET, and also be studied at Mu2e.

Coulomb bound states of positively charged muons and electrons ($\mu^+e^-$) can also be formed: Muonium (Mu) [19] atoms also offer the possibility to look for NP, for example the cLFV Muonium-antimuonium conversion [20], or muonium decay to an electron-positron pair $\mathrm{Mu} \to e^-e^-$. The latter has not been experimentally searched for yet; concerning the former, PSI has put upper bounds on the conversion probability P(Mu-$\overline{\mathrm{Mu}}$)$< 8.3 \times 10^{-11}$ [21].

Due to its comparatively large mass, tau leptons decay via numerous leptonic and semi-leptonic modes, several of them violating charged lepton flavour. Figure 1 summarises the current bounds on $\tau$ cLFV decay modes, as reported by the HFLAV Collaboration [22, 23]. In the future, Belle II is expected to improve these bounds by 1-2 orders of magnitude [24].

Meson decays also offer excellent testing grounds for cLFV - the sensitivity to several

---

[1]For dedicated reviews see, for example, [2–5].

[2]For a recent discussion of spin-dependent contributions to $\mu-e$ conversion, see [11].

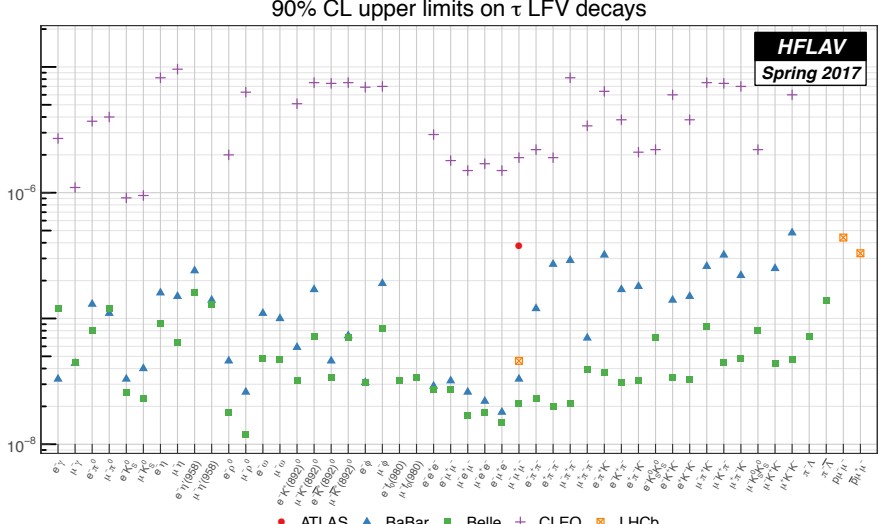

Figure 1: HFLAV summar plot on the upper limits for tau lepton-flavor-violating branching fractions. From [22].

modes is already very competitive, especially with $\mu e$ final states. For example, one has [1] $\mathrm{BR}(K_L \to \mu e) < 4.7 \times 10^{-12}$, $\mathrm{BR}(K^+ \to \pi^+ \mu^+ e^-) < 2.1 \times 10^{-11}$, $\mathrm{BR}(D^0 \to \mu e) < 1.5 \times 10^{-8}$ $\mathrm{BR}(B \to \mu e) < 2.8 \times 10^{-9}$, among many others.

At higher energies one can also look for the cLFV decays of SM neutral bosons, $Z \to \ell_i \ell_j$ and $H \to \ell_i \ell_j$. Data from LEP and the LHC has allowed to constrain several of the above decays [1]: $\mathrm{BR}(Z \to e\mu) < 7.5 \times 10^{-7}$, $\mathrm{BR}(Z \to \mu\tau) < 1.2 \times 10^{-5}$, and $\mathrm{BR}(Z \to e\tau) < 9.8 \times 10^{-6}$; $\mathrm{BR}(H \to e\mu) \lesssim 3.5 \times 10^{-4}$ $\mathrm{BR}(H \to \mu\tau) \lesssim 1.43\%$ and $\mathrm{BR}(H \to e\tau) \lesssim 0.0069$.

## 2.2 LNV processes

The quest for the violation of total lepton number is especially appealing concerning New Physics models of neutrino mass generation, since several of the latter call upon the addition of new neutral Majorana fermions to the SM content. LNV processes (in general $\Delta L = 2, 4$ transitions) have been extensively searched for, and are an important part of the experimental neutrino programme and of high-intensity searches.

One of the best known observables is neutrinoless double beta decay ($0\nu 2\beta$), corresponding to a $\Delta L = 2$ transition, $(A, Z) \to (A, Z + 2) + 2e^-$. At present, the best bounds on the electron effective mass have been determined by the KamLAND-Zen Collaboration, $m_{ee} \lesssim (61\text{-}165)$ meV [25]. Several collaborations are dedicated to searching for these transitions, with future sensitivities on the ballpark of 50 meV.

Muonic atoms can also undergo LNV decays: $\mu^- + (A, Z) \to e^+ + (A, Z - 2)^*$, in which the asterisk denotes the fact that the final state nucleus can be in an excited state. Current bounds have been obtained by the SINDRUM Collaboration [26], $\mathrm{CR}(\mu^- + \mathrm{Ti} \to e^+ + \mathrm{Ca}^{(*)}) \lesssim 3.6 \times 10^{-11} (1.7 \times 10^{-12})$, and the process could also be experimentally searched for at COMET and Mu2e.

Leptonic and semileptonic meson and tau decays also include many LNV modes; in Table 1 we briefly illustrate the bounds on a few of the channels.

Table 1: Current bounds on a small set of LNV semileptonic meson and tau decays [1, 27, 28].

| Meson LNV decay | Current Bound | |
|---|---|---|
| | $\ell = e,\ \ell' = e$ | $\ell = \mu,\ \ell' = \mu$ |
| $K^- \to \ell^- \ell'^- \pi^+$ | $6.4 \times 10^{-10}$ | $1.1 \times 10^{-9}$ |
| $D^- \to \ell^- \ell'^- \pi^+$ | $1.1 \times 10^{-6}$ | $2.2 \times 10^{-8}$ |
| $D^- \to \ell^- \ell'^- K^+$ | $9.0 \times 10^{-7}$ | $1.0 \times 10^{-5}$ |
| $B^- \to \ell^- \ell'^- \pi^+$ | $2.3 \times 10^{-8}$ | $4.0 \times 10^{-9}$ |
| $B^- \to \ell^- \ell'^- K^+$ | $3.0 \times 10^{-8}$ | $4.1 \times 10^{-8}$ |
| $B^- \to \ell^- \ell'^- \rho^+$ | $1.7 \times 10^{-7}$ | $4.2 \times 10^{-7}$ |
| $B^- \to \ell^- \ell'^- D^+$ | $2.6 \times 10^{-6}$ | $6.9 \times 10^{-7}$ |

| $\tau$ LNV decay | Current Bound | |
|---|---|---|
| | $\ell = e$ | $\ell = \mu$ |
| $\tau^- \to \ell^+ \pi^- \pi^-$ | $2.0 \times 10^{-8}$ | $3.9 \times 10^{-8}$ |
| $\tau^- \to \ell^+ \pi^- K^-$ | $3.2 \times 10^{-8}$ | $4.8 \times 10^{-8}$ |
| $\tau^- \to \ell^+ K^- K^-$ | $3.3 \times 10^{-8}$ | $4.7 \times 10^{-8}$ |

## 3 Leptonic observables and New Physics models

The interpretation of experimental data, and the possibility of inferring constraints on the SM extension that could be at their origin, requires considering concrete theoretical frameworks. As extensively discussed by A. Signer in his contribution[3], experimental bounds can be used to successfully constrain NP contributions in a model-independent way, by means of the effective approach. Here we will focus on a model-dependent approach, in particular on NP models accounting for neutrino masses and leptonic mixings, and explore to which extent cLFV and LNV data can allow to probe these realisations. However, it is important to highlight that despite it being a very appealing possibility, cLFV (and LNV) observables need not be associated with a mechanism of neutrino mass generation. Several NP models include explicit sources of flavour violation, which can lead to contributions to many of the observables discussed in the previous section. Moreover, and even if the SM extension does include a mechanism of neutrino mass generation, the NP scale and states responsible for $m_\nu$ are not necessarily those involved in cLFV transitions and decays. In terms of the effective approach, and casting the effective Lagrangian as

$$\mathcal{L}^{\text{eff}} = \mathcal{L}^{\text{SM}} + \sum_{n \geq 5} \frac{1}{\Lambda^{n-4}} \mathcal{C}^n \mathcal{O}^n , \tag{1}$$

in which $\Lambda$ denotes a high-scale associated with NP, and $\mathcal{C}^n$, $\mathcal{O}^n$ are the higher-dimensional effective couplings and operators, the above discussion corresponds to having independent contributions to the dimension 5 (Weinberg) operator (responsible for neutrino mass generation) and dimension 6 operators (among which those responsible for cLFV transitions). Furthermore, the associated scales can be different, i.e. $\Lambda_{\text{LNV}}^{(5)} \neq \Lambda_{\text{cLFV}}^{(6)}$.

In the following, we thus carry a very brief survey of some illustrative examples of NP models of massive neutrinos, discussing their impact for cLFV and LNV observables. Interestingly, many models of neutrino mass generation call upon the presence of sterile neutrinos (singlets under the SM gauge group), which are by themselves well-motivated NP candidates. A useful first approach to address cLFV and LNV in the presence of massive neutrinos is to consider simple ad hoc constructions in which the SM is extended by one (or more) massive sterile

---

[3]See A. Signer, *Proceedings for the 15th International Workshop on Tau Lepton Physics, Amsterdam, The Netherlands, 24-28 September 2018* .

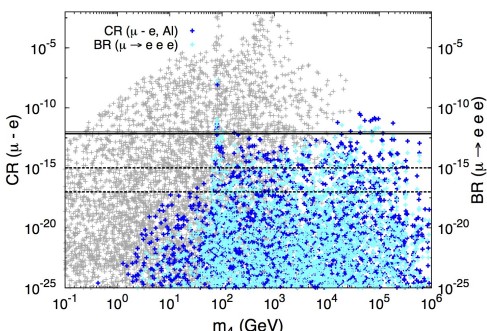
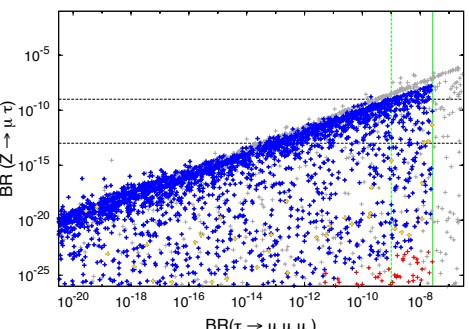

Figure 2: On the left, CR($\mu - e$) and BR($\mu \rightarrow eee$) as a function of $m_4$; the former is displayed in dark blue (left axis), while the latter is depicted in cyan (right axis). A thick (thin) solid horizontal line denotes the current experimental bound on the CR($\mu - e$, Au) [13] ($\mu \rightarrow eee$ decays [9]), while dashed lines correspond to future sensitivities to CR($\mu - e$, N) [15, 29] (from [30]). On the right, BR($Z \rightarrow \tau\mu$) vs. BR($\tau \rightarrow 3\mu$); vertical lines denote future experimental sensitivities while the horizontal ones correspond to the prospects of a GigaZ facility and of the FCC-ee (from [31]). Both cases correspond to a "3+1 toy model", and in both panels grey points are phenomenologically excluded.

fermion, whose effects encode those of a larger set of NP states that could be present.

## 3.1 Simple "toy models"

As mentioned above, the simple "toy models" consist in a phenomenological bottom-up approach, without any formal assumption on the underlying mechanism of mass generation. In practical terms, one considers that leptonic mixings and neutrino masses are not correlated; for the case of an additional Majorana sterile state, the "3+1 toy model" is described by the masses of the light (active) neutrinos and that of the heavy fermion, and by a $4 \times 4$ unitary mixing matrix that relates the physical and the interaction bases, **U**. The mixing matrix is parametrised by 6 mixing angles and 6 CP violating phases, and its upper non-unitary $3 \times 3$ sub-block now encodes left-handed leptonic mixings (the would-be $U_{\text{PMNS}}$). This leads to the modification of charged and neutral lepton currents, and translates into contributions to several cLFV and LNV observales, as we proceed to discuss.

In order to illustrate the phenomenological prospects of such a minimal SM extension in what concerns cLFV, we display in the left panel of Figure 2 the expected contributions to two cLFV muon channels: the neutrinoless muon-electron conversion in nuclei (on the left axis) and the muon decay into 3 electrons (on the right axis), both observables shown as a function of the mass of the heavy sterile state, $m_4$. The contributions can be very large, especially for sterile masses above the electroweak threshold, and lie well within the sensitivity of future dedicated facilities (Mu2e/COMET and Mu3e).

The sizeable contributions to both these observables also indirectly preclude the observability of radiative muon decays, as the branching ratios which would lie within MEG II sensitivity are already excluded due to violation of current limits on $\mu \rightarrow 3e$ and $\mu - e$ conversion. For heavy states on the mass regime associated with the largest rates, one verifies that the 3-body decays (and $\mu - e$ conversion) receive the dominant contributions from $Z$-penguins, so that it is only natural to expect that there will be a strong correlation between lepton flavour violating $Z \rightarrow \ell_i \ell_j$ decays and low-energy cLFV observables, such as $\ell_i \rightarrow 3\ell_j$. This is indeed the case,

as shown on the right panel of Fig. 2, in which we display the correlation of BR($Z \to \mu\tau$) vs. BR($\tau \to 3\mu$). Firstly, let us notice that one can expect sizeable rates for the cLFV $Z \to \mu\tau$ decays, potentially observable at a future FCC-ee [31]; the correlation between the two observables is also manifest. Finally, it is important to stress that not only cLFV $Z$ decays can probe flavour violation in the $\mu - \tau$ sector beyond the reach of Belle II, but that an important region of the parameter can be experimentally probed in a 3-fold way: at high-energies (via $Z$ decays), at high-intensities ($\tau \to 3\mu$), and also in searches for $0\nu2\beta$ decays.

Concerning the latter point, it is worthwhile to emphasise that the additional sterile state can also modify the predictions to LNV neutrinoless double beta decays (see, e.g. [32, 33]), leading to an augmented range of the corresponding predictions - which are within future experimental sensitivity, even for a normal ordering of the light neutrino spectrum.

Additional sterile states can also have a significant impact for LNV observables (other than $0\nu2\beta$ decays); if the sterile states have a mass allowing them to be produced on-shell from meson and tau decays, then - and should they be of Majorana nature - one can have a resonant enhancement of the rates for LNV semileptonic meson and tau decays [34, 35]. Relying on the abundant experimental data on these decays, a full update was recently carried to evaluate the new constraints that the LNV decay modes imply for the sterile neutrino parameter space [35]. As visible from the left panel of Fig. 3, in which we display the BR($\tau^- \to \mu^+ P_1^- P_2^-$) - with $P_i^-$ light pseudoscalar mesons - as a function of the mass of the mediating Majorana sterile state, several modes can have large branching fractions, some even already in conflict with experimental bounds. Furthermore, the extensive data obtained from the vast array of decay modes studied allowed to infer bounds on all the entries of a (generalised) effective Majorana mass matrix [35],

$$
m_\nu^{\ell_\alpha \ell_\beta} = \left| \sum_{i=1}^{4} \frac{U_{\alpha i}\, m_i\, U_{\beta i}}{1 - m_i^2/p_{12}^2 + i\, m_i \Gamma_i/p_{12}^2} \right|,
\tag{2}
$$

in which $U_{\alpha i}$ denote the elements of the $4 \times 4$ mixing matrix, $m_i\,(\Gamma_i)$ the neutrino masses (widths) and $p_{12}^2$ the momentum transfer. While the $ee$ entry clearly receives the best bounds from $0\nu2\beta$ decays, improved bounds were established for the remaining five independent elements, typically below $10^{-3}$ GeV [35]. This is illustrated, for the "3+1 toy model", on the right panel of Fig. 3, in which one finds the allowed range for $|m_\nu^{\mu\tau}|$ versus the branching ratio of the LNV decay which provides the best constraints, in this case, $B^+ \to \mu^+ \tau^+ \pi^-$. The colour code denotes the mass regime of the sterile state mediating the process.

## 3.2 Seesaw realisations

Although cLFV need not arise from the underlying NP model responsible for neutrino mass generation, models in which this is indeed the case - such as the different seesaw realisations - are particularly appealing frameworks. While theoretically appealing, realisations of the seesaw at very large scale (high-scale seesaws) have little impact on the cLFV observables here discussed: despite the associated large "natural" values for the neutrino Yukawa couplings, the contributions to the rates are heavily suppressed due to the very large mass of the mediators. Low-scale seesaw realisations, in which the comparatively light new states (with masses between the MeV and the TeV) have non-negligible mixings with the active states and hence do not decouple, offer very rich prospects for both cLFV and LNV observables. This is illustrated on the left panel of Fig. 4, which displays the predictions to several cLFV muon channels as a function of the mass of the mediators, for a low-scale realisation of a type I seesaw [36]. For states with masses above the tenths of GeV, one expects sizeable contributions, well within experimental sensitivity.

Another theoretically appealing realisation is the "Inverse Seesaw" (ISS) [37–39], in which the SM content is enlarged by several generations of sterile states, in addition to right-handed

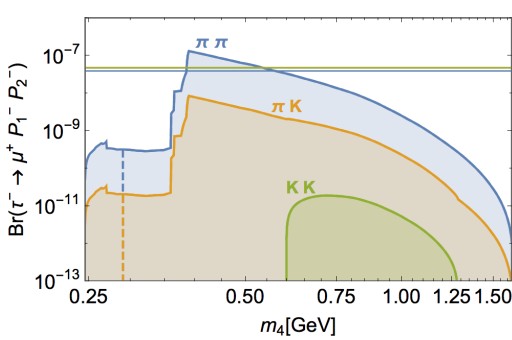
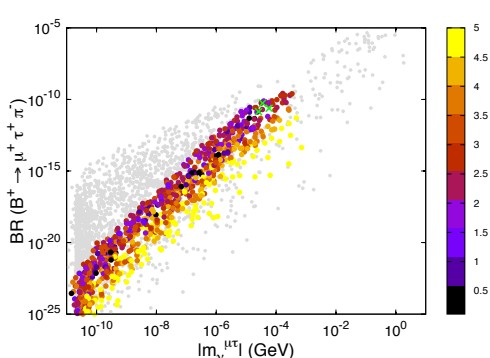

Figure 3: On the left, LNV BR($\tau^- \to \mu^+ P_1^- P_2^-$), with $(P_1^- P_2^-) = (\pi^- \pi^-, \pi^- K^-, K^- K^-)$, as a function of the heavy sterile state mass, $m_4$. Pale blue, yellow and green curves (surfaces) respectively denote the maximal (allowed) values of the corresponding BRs. On the right, BR($B^+ \to \mu^+ \tau^+ \pi^-$) as a function of $|m_\nu^{\mu\tau}|$ (in GeV). The colour code denotes the mass regime of the sterile state mediating the process ($m_4$, in GeV); grey points denote exclusions due to the violation of experimental/observational bounds. Both panels correspond to a "3+1 toy model". From [35].

neutrinos (three of each in the (3,3) ISS realisation). The smallness of the light neutrino masses is ensured by an approximate conservation of total lepton number; the heavier states combine to form 3 pseudo-Dirac pairs. In what concerns contributions to the cLFV observables previously discussed, the ISS is associated with sizeable contributions to several modes; for example, $\mu \to 3e$ decays, neutrinoless conversion in nuclei, and $Z \to \mu\tau$ decays receive sizeable contributions, potentially within future sensitivity [30,31]. On the other hand, $\tau \to 3\mu$ clearly lies beyond the reach of Belle II, as the regimes corresponding to large values of the branching ratio are precluded due to leading to the violation of other (cLFV) bounds. This is visible from the right panel of Fig. 4.

Having the heavy states forming pseudo-Dirac pairs leads to a suppression of LNV rates (including the semileptonic meson and tau lepton decays discussed in the previous subsection).

At high-energy colliders, the ISS can also be at the origin of interesting cLFV signatures, as is the case of the very clean channel leading to a final state composed of $\mu\tau$ pairs and 2 jets (no missing energy). As shown in [40], one can expect a significant number of events after cuts.

The non-singlet seesaws (i.e., the type II and III realisations) are associated with very distinctive cLFV signatures, a direct consequence of the triplet nature of the mediators. While in the type I seesaw (and its variants) all cLFV processes occur at the loop level, in the type II seesaw 3-body decays occur at tree-level; in the type III, all observables - with the exception of radiative decays - are tree-level processes. In the case of a future observation, constructing ratios of observables may be a powerful means to disentangle the different realisations: as an illustrative example, one has BR($\mu \to e\gamma$)/BR($\mu \to 3e$)$\approx 10^{-3}$ (1) for type III (I); likewise one finds CR($\mu - e$, Ti)/BR($\mu \to e\gamma$)$\approx 10^3$ ([0.05 − 5]) for type III (II) [41].

### 3.3 Further NP models: additional fields, symmetries and complete frameworks

With the goal of addressing other observational and theoretical problems of the SM - in addition to neutrino mass generation - several complete models of NP offer very rich prospects for

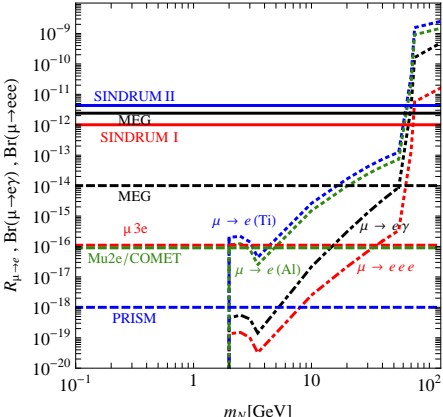
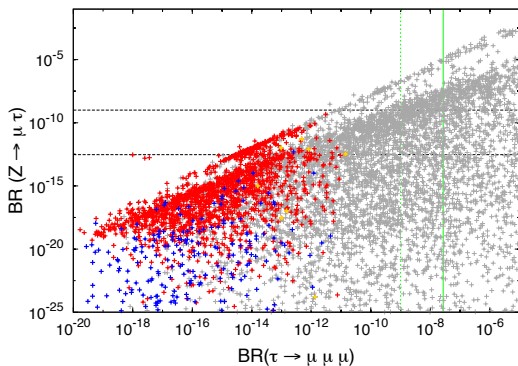

Figure 4: On the left, predictions obtained in a low-scale type I seesaw for the maximal allowed cLFV rates compatible with current searches; horizontal full (dashed) lines denote present (future) experimental sensitivity. From [36]. On the right, a (3,3) ISS realisation: BR($Z \to \tau\mu$) vs. BR($\tau \to 3\mu$); vertical lines denote future experimental sensitivities while the horizontal ones correspond to the prospects of a GigaZ facility and of the FCC-ee; grey points are phenomenologically excluded. From [30].

cLFV (and LNV).

In view of their unique potential to address the $B$-meson decay anomalies, SM extensions via leptoquarks (scalar or vector fields) have been receiving increasing attention. By construction, these models break lepton flavour universality, and many have a strong impact to flavour violating observables, both in the quark and lepton sectors. As an example, a SM leptoquark extension which aims at simultaneously addressing the $R_{K^{(*)}}$ anomalies, account for neutrino oscillation data and put forward a viable DM candidate has been recently considered [42]: 2 scalar leptoquarks and three generations of lepton triplets are added to the SM, whose gauge group is reinforced by a $Z_2$ symmetry. This is also an example of a model in which neutrino masses are radiatively generated at the 3-loop level. While the model can indeed account for $R_{K^{(*)}}$, and a viable DM relic density, the parameter space is strongly constrained from flavour observables, the most stringent ones being $\mu - e$ conversion and $K \to \pi\nu\nu$ decays [42]. This is illustrated by the left panel of Fig. 5, where the distinct BRs are depicted as a function of the mass of the $h_1$ leptoquark.

Models leading to the restoration of parity in SM gauge interactions are also well-motivated and appealing NP constructions; naturally including right-handed neutrinos - as well as new gauge bosons and bidoublet and triplet Higgs -, Left-Right (LR) symmetric models automatically incorporate a hybrid type I-II seesaw mechanism. Many realisations (see, e.g. [44, 45]) lead to extensive contributions to high-intensity and high-energy cLFV and LNV observables, which further exhibit a high degree of correlation.

In its different realisations, the seesaw can also be embedded in the framework of otherwise flavour conserving supersymmetric (SUSY) extensions of the SM; SUSY models in which the neutrino Yukawa couplings are the unique source of (lepton) flavour violation leads to a high degree of correlation between all cLFV observables, both at low- and high-energies[4]. One

---

[4]This is also an interesting example of models in which the scale of neutrino mass generation - the seesaw scale, close to the Grand Unification scale - is very different from that of cLFV, associated with the SUSY propagators. Following the discussion at the beginnng of this section, one has $\Lambda^{(5)}_{\mathrm{LNV}} \gg \Lambda^{(6)}_{\mathrm{cLFV}}$.

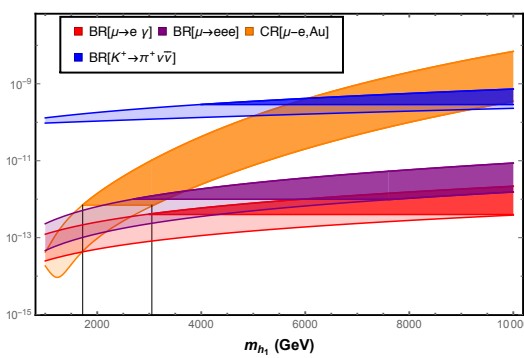
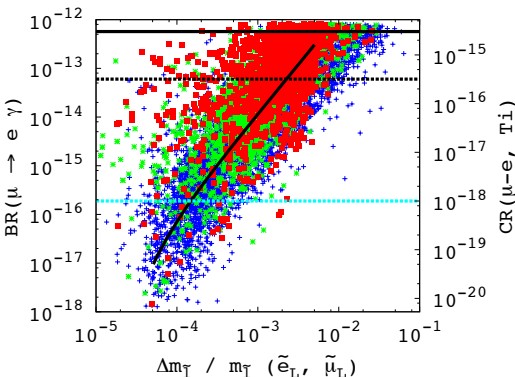

Figure 5: On the left, contributions to BR($\mu \to e\gamma$), BR($\mu \to 3e$), CR($\mu - e$, Au) and BR($K^+ \to \pi^+ \nu\bar{\nu}$) as a function of the $h_1$ leptoquark mass $m_{h_1}$, for $h_1$ flavour couplings complying with the current intervals for $R_{K^{(*)}}$; light (dark) surfaces denote currently allowed (excluded) regimes due to the violation of the associated experimental bound (from [42]). On the right panel, 1$^{\text{st}}$ and 2$^{\text{nd}}$ generation charged slepton mass splittings vs. BR($\mu \to e\gamma$), with CR($\mu - e$, Ti) on the secondary y-axis in a type I SUSY seesaw, for different values of the heaviest right-handed neutrino mass $M_{R_3} = 10^{13,14,15}$ GeV ($M_{R_1, R_2} = 10^{10,11}$ GeV) and for a flavour conserving modified mSUGRA benchmark (from [43]).

such example is shown on the right panel of Fig. 5, in which the vertical axes correspond to low-energy cLFV observables ($\mu \to e\gamma$ and $\mu - e$ conversion), displayed as a function of the relative mass splittings between left-handed smuons and selectrons - a high-energy cLFV observable [43]. The strong synergy of these observables can be explored to falsify the model, or then allow to infer hints in the scale of the seesaw mediators - which would be otherwise unreachable.

Finally, one can explore Grand Unified (GUT) models, appealing theoretical constructions which succeed in reducing the arbitrariness of the Yukawa couplings, establishing links between the lepton and quark flavour observables. Although dependent on the specific model, GUTs in general lead to predictive scenarios, not only concerning neutrino mass generation, but also to interesting correlation patterns between many cLFV observables.

## 4  Concluding remarks

The observation of neutrino oscillations confirmed that lepton flavour was violated in the neutral lepton sector, thus providing the first laboratory evidence for physics beyond the Standard Model; currently, a number of tensions between experimental data and SM expectations - all nested in lepton related-observables - further suggests the need to consider NP scenarios. At present, there is a massive, world-wide experimental effort focused on searching for cLFV and LNV rare decays and transitions, among which the observables that were briefly reviewed here. The observation of any of these processes would constitute a clear signal of New Physics - beyond the SM extended via massive (Dirac) neutrinos.

Although this need not be the case, a very appealing hypothesis is that the NP model responsible for the observables here discussed, is also associated with the mechanism of neutrino mass generation. We have considered the prospects of several SM extensions via massive neutrinos (from simple toy models to complete constructions) in what concerns cLFV and LNV. Clearly, the synergy between direct searches at high-energy colliders, $\nu$ physics and

high-intensity observables must be fully explored to probe, constrain and possibly unveil the underlying New Physics model.

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
