# Peer review of "Lepton flavour violation and neutrino masses"

_SciPost Physics Proceedings, doi:SciPost Phys. Proc. 1, 015 (2019)_

## Round 1 · Referee Report · Anonymous (Referee 1) · 2018-12-12

Report
The contribution provides a review about the possible new phenomena that one can expect from the established phenomenon of neutrino oscillations, per se necessitating an extension of the SM. These new phenomena notably include, among the others, charged-lepton flavour violation, lepton number violation and contributions to lepton dipole moments.
The contribution starts from a review of the experimental status of such searches, and proceeds with a reasoned discussion of their potential within chosen SM extensions. Given the fact that flavour observables are indirect probes of a given SM extension, crucial is the interplay among as many observables as possible. The discussion focuses precisely on this aspect.
The contribution is carefully written, and an enjoyable read. I have one minor comment, concerning fig. 5 (left). The caption mentions "light (dark) surfaces denote currently allowed (excluded) regimes due to the violation of the associated experimental bound", but I can't see such surfaces in the plot. In the original reference, such surfaces seem to be visible only in a different plot, not shown in this contribution. I also suppose that the two vertical lines on fig. 5 (left) denote the LQ-mass region explaining RK(*). If so, I would specify this in the caption.
The contribution starts from a review of the experimental status of such searches, and proceeds with a reasoned discussion of their potential within chosen SM extensions. Given the fact that flavour observables are indirect probes of a given SM extension, crucial is the interplay among as many observables as possible. The discussion focuses precisely on this aspect.
The contribution is carefully written, and an enjoyable read. I have one minor comment, concerning fig. 5 (left). The caption mentions "light (dark) surfaces denote currently allowed (excluded) regimes due to the violation of the associated experimental bound", but I can't see such surfaces in the plot. In the original reference, such surfaces seem to be visible only in a different plot, not shown in this contribution. I also suppose that the two vertical lines on fig. 5 (left) denote the LQ-mass region explaining RK(*). If so, I would specify this in the caption.

Author: Ana M. Teixeira on 2018-12-13 [id 380]
(in reply to Report 1 on 2018-12-12)Dear Editor, dear Referee,
thank you very much for the report.
Concerning the points raised in relation with Fig. 5 (left), I would like to clarify that there are indeed light /dark surfaces : these correspond to light/dark blue, light/dark yellow (red, violet), which are indeed present in this plot. Regarding the vertical lines, they are "lengthier" to explain (corresponding to the final regimes of the LQ masses for which all the experimental constraints are verified), and since I made no reference to them neither in the presentation nor in the proceedings, I chose not to refer to them. Of course these are described in the original paper. My goal in the presentation/proceedings was to highlight the constraining power of the cLFV observables. Should the Referee feel that edition is indeed required, I will then accordingly do so.
I hope that this clarifies the points raised by the Referee, and that the contribution can be accepted for publication.
Cordially yours,
Ana M. Teixeira

---

## Editorial Decision

published